# Age Group Classification of Dental Radiography without Precise Age Information Using Convolutional Neural Networks

**DOI:** 10.3390/healthcare11081068

**Published:** 2023-04-08

**Authors:** Yu-Rin Kim, Jae-Hyeok Choi, Jihyeong Ko, Young-Jin Jung, Byeongjun Kim, Seoul-Hee Nam, Won-Du Chang

**Affiliations:** 1Department of Dental Hygiene, Silla University, 140 Baegyang-daero 700 Beon-gil, Sasang-gu, Busan 46958, Republic of Korea; 2Department of Artificial Intelligence, Pukyong National University, Busan 48513, Republic of Korea; 3Department of Biomedical Engineering, Chonnam National University, Yeosu 59626, Republic of Korea; 4School of Healthcare and Biomedical Engineering, Chonnam National University, Yeosu 59626, Republic of Korea; 5Department of Dental Hygiene, Kangwon National University, Samcheok 25913, Republic of Korea

**Keywords:** dental age estimation, forensic dentistry, deep learning, oral health, data augmentation

## Abstract

Automatic age estimation using panoramic dental radiographic images is an important procedure for forensics and personal oral healthcare. The accuracies of the age estimation have increased recently with the advances in deep neural networks (DNN), but DNN requires large sizes of the labeled dataset which is not always available. This study examined whether a deep neural network is able to estimate tooth ages when precise age information is not given. A deep neural network model was developed and applied to age estimation using an image augmentation technique. A total of 10,023 original images were classified according to age groups (in decades, from the 10s to the 70s). The proposed model was validated using a 10-fold cross-validation technique for precise evaluation, and the accuracies of the predicted tooth ages were calculated by varying the tolerance. The accuracies were 53.846% with a tolerance of ±5 years, 95.121% with ±15 years, and 99.581% with ±25 years, which means the probability for the estimation error to be larger than one age group is 0.419%. The results indicate that artificial intelligence has potential not only in the forensic aspect but also in the clinical aspect of oral care.

## 1. Introduction

In human civilization, faces and hands have their own individuality and have been extensively studied sociobiologically. In particular, the teeth and bone skeleton of the craniofacial region are generally the best-preserved parts in humans, and individual identification is possible because the size, shape, and proportion, as well as the results of surgery and treatment, are different [1]. In addition, because the jawbone and teeth differ according to the growth period, they have been used for age estimation. In particular, teeth are used as reliable data when estimating age because they are the longest preserved among human tissues and change according to age is relatively gradual [2]. The age estimation of teeth plays an important role in clinical and forensic science. It is used for criminal responsibility investigations of living individuals, such as in cases of large-scale disasters, fires, accidents, and murders [3,4].

Demirjian’s “eight stages” is the most commonly used method for estimating age from oral anatomical structures [5,6]. This is a very simple method for scoring bone calcification and maturity, but it has limitations that apply to children and adolescents. A new scoring method that corrects and supplements this has been proposed, but it is unsuitable because of the high error in age estimation resulting from the subjective judgment of the observer; therefore, a study on a more accurate method is needed [7,8]. According to Caggiano et al. [6], as a result of a study using third molar radiographs in a population in southern Italy, the accuracy and reproducibility of the Demirjian method were confirmed over 90%, but it was suggested that additional research on a larger study population is needed. Recently, an age estimation method based on the methylation level of DNA extracted from teeth using a real-time methylation-specific polymerase chain reaction was reported as evidence that DNA methylation in the human genome isolated from bodily fluids changes with age [9]. However, this method has a clear limitation in that age estimation is possible only when the tooth is extracted posthumously. In contrast to other forensic age estimation methods, the radiological method is accessible to clinicians, relatively simple, and non-destructive [10]. It has the advantage that it can be applied to living people because it can be estimated by the decrease and change in the size of the pulp cavity with increasing age [11]. According to a study by Kvaal et al. [12], the method of estimating age by measuring the dimensions and length and width of teeth on apical radiographs has been suggested as a representative method using radiographs for age estimation with relatively high accuracy.

In 2015, a method for estimating age using panoramic radiographs, commonly used in dentistry, was proposed [13], and much research on age estimation based on artificial intelligence (AI) has been conducted [14,15,16]. The reason for using the panorama is that it is basic radiography for diagnostic and forensic medical treatment in dentistry, and the data derived from the panorama are highly reliable [17]. Therefore, AI using panoramas has been reported to be more accurate than traditional radiation methods because it can predict tooth age more accurately and efficiently through machine learning [14,15,16]. Galibourg [14] and Tao [15] applied machine learning to the existing scoring method for age estimation; however, this method still had a large error owing to the subjective judgment of the observer. Accordingly, attempts have recently been made to estimate age without human intervention using convolutional neural networks (CNNs) [18,19,20]. CNNs have been used to diagnose diseases such as breast cancer [21], skin cancer [22], diabetic retinopathy [23], dental caries [24], and periodontal disease [25], as well as age estimations. However, research on their clinical aspects is still lacking. Researchers investigating age estimation using panoramas have reported the high usability of CNNs; however, the number of learned panoramas was remarkably small and biased toward younger age groups, suggesting the need for additional research. CNNs generally require huge datasets, but collecting images with precise age information is challenging.

In this study, an attempt was made to estimate ages based on the median age of the subjects by using a larger number of panoramic images from various age groups. Therefore, this study was conducted to prove that precise estimation of tooth age is possible even without precise age information. We designed a convolutional neural network for this purpose and utilized it to estimate the approximate age of the teeth images.

The remainder of this paper is organized as follows. Section 2 describes the research method by explaining the data and network models. Section 3 presents the experimental results, and the discussions follow in Section 4. Finally, Section 5 gives conclusions.

## 2. Materials and Methods

A total of 10,023 dental panoramic images were collected by the Institutional Review Board of Kangwon National University, Republic of Korea. The images were grouped into seven categories according to age (in decades from the 10s to the 70s). The original panoramic images were 1504 × 2768 pixels in size and included other parts of the face, such as jaws and noses. To facilitate learning, the tooth regions were cropped and resized into 256 × 512. The images were augmented by rotating them through −0.2 to 0.2 rad, and they were flipped horizontally to increase the stability of the model.

A network model was designed to estimate the age of the teeth in the cropped image, as shown in Figure 1. The model starts with the preprocessing layer of the random flips and rotations to augment a small number of teeth images. We used five sets of convolutional and max-pooling layers, a flatten and dropout layer, and a dense and dropout layer, as shown in Figure 1. Convolutional layers were attached to the preprocessing layer to extract spatial features. The five convolutional layers were utilized with different numbers of filters (8, 16, 32, 64, and 128 in series), and the max-pooling layer with a pool size of (2,2) was attached to the convolutional layer. The extracted features were then converted into the one-dimensional form using the flatten layer, and dental age was estimated using the fully connected layers. The dropout layers were attached to the flatten and fully connected layers.

In this study, k-fold cross-validation was employed to verify the accuracy. Generally, it is used with a small quantity of image data. As shown in Figure 2, k was set to 10. The data were randomly divided into 10 folds (groups), and the training and testing were performed 10 times by changing the test data. This means that the 10th fold was used for the test data, and the rest were used for training. Then, the ninth to first folds were utilized one by one for testing. The final errors or accuracies were calculated by averaging the results of the ten folds.

The target value for each age group was set to the median age, i.e., the target values for ages in the 10s, 20s, 30s, 40s, 50s, 60s, and 70s were 15, 25, 35, 45, 55, 65, and 75, respectively. This was because precise age information was not recorded during data acquisition.

Therefore, the concept of tolerance must be employed (see Figure 3). For a tolerance age of ±5 years, a dental image that is in the 30s group and estimated at 39.5 years is considered to be estimated accurately because the median value of the 30s is 35. When the tolerance range is ±15 years, the confusion about whether an image should be assigned to a particular age group or its neighboring group is accepted. In other words, a dental image of an individual in their 30s and estimated to be 49.5 years old is considered an accurate estimate.

## 3. Results

The estimated ages and the corresponding actual ages are shown in the box plot in Figure 4. The lines in the boxes represent the median values of the predicted ages. The vertical lines connected to the boxes represent the range from the minimum to the maximum predicted age. Each colored box contains 25–75% of the predicted age values. Diamond marks indicate abnormal values. The figure shows that the box of 25–50% of the predicted ages was approximately within the target age group except for the 70s.

The numbers of images with errors are shown according to error size in Figure 5. The numbers of error images were 4626 with a tolerance of ±5 years, 489 with a tolerance of 15 years, and 42 with a tolerance of 25 years.

Figure 6 shows the relationship between accuracy and tolerance in dental age prediction. The accuracy increased as the tolerance age increased. The accuracy was 53.846% with a tolerance of ±5 and 95.121% with a tolerance of ±15. The accuracy with a tolerance of ±25 was 99.581% (Table 1).

Figure 7 shows a confusion matrix of the estimation results. Clearly, the most confusion occurred between the adjusted groups (10s and 20s, 20s and 30s, and so on). The ages were often overestimated for ages in the 40s and 50s: 49 and 94 images of the 40s and 50s age groups were classified as being in the 60s and 70s, respectively, whereas images for which age was underestimated were fewer (32 and 55 images for the age groups of the 40s and 50s, respectively). One reason was that the teeth of persons in their 40s and 50s have frequently been treated extensively or removed. Confusions between the adjacent age groups occurred frequently for the 60s, as 54% of teeth images were misclassified to 50s or 70s. The confusions of 10s or 70s were relatively lower 13.11% and 25.14% of 10s or 70s were misclassified to their adjacent age groups.

Figure 8 shows examples of correctly classified images. In this study, 5397 of 10,023 dental panoramic radiographs were predicted successfully within a 5-year tolerance. Figure 8a–d show the radiographs of four persons in their 70s, 50s, 10s, and 70s, respectively, and their ages were predicted to be 76.89, 57.89, 16.22, and 73.20 years, respectively.

The reasons for the misclassification of the images, especially when the age errors were smaller than 15 years, could not be determined. No significant differences were found between the misclassified images and the accurately classified images. Figure 9 shows an example. The actual age of the subject whose radiograph is shown in the figure was in the 20s; however, it was predicted to be 30.61 years.

The number of images with an estimated age error of more than ±15 years was 489 out of 10,023. A few examples are shown in Figure 10. Figure 10a–d show the radiographs of subjects in their 50s, 30s, 70s, and 30s, respectively, whose ages were predicted to be 38.31, 55.07, 59.41, and 70.09 years, respectively. As shown, the underestimation or overestimation of age seems to be highly related to implants or the number of treated teeth.

The number of images with estimated age errors of more than ±25 years was 42 out of 10,023. Figure 11 shows examples. Figure 11a,b,d show the radiographs of subjects in their 20s, whose ages were predicted to be 67.22, 69.52, and 51.78 years, respectively. Many of the subjects’ teeth in these images had been removed (Figure 11b,d) or decayed (Figure 11a). The image shown in Figure 11c is of a subject in the 20s group, whose age was predicted to be 61.59 years. This case is slightly different from the others in that the subject had a single implant but only a few treated teeth. The age classification may have been misclassified because many teeth were lost and the alveolar bone was lowered in the image.

## 4. Discussion

Age is a predictor of physical, emotional, and social development and maturation. Chronological age is an objective indicator of age and is simply the age at which elapsed time is calculated according to a calendar. This is commonly considered the “actual age”. However, biological age represents an individual’s level of biological and physiological development, maturation, and physical health, and it can be lower or greater than the actual age because it is determined by the individual’s current position in the lifespan [26]. Oral age is calculated as an objectively measurable oral health index and is a biological age that reflects oral health status. Oral age reflects the development of modern medicine and lifestyle changes and can be considered a concept related to biological age that can be compared based on the average oral health status of an entire nation [27,28]. It is an age estimation method that is widely used to estimate an accurate age. It is a widely used age estimation method for estimating an accurate age, and the teeth are preserved for a long time and can be observed directly. It is also useful for age estimation because it is known that individual differences are the least due to relatively gradual changes according to age [29]. Accordingly, the American Society of Forensic Odontology and the Study Group on Forensic Age Diagnostics recommended radiographic dental age estimation to help estimate chronological age [30,31].

Among intraoral radiographs, panoramic radiographs have the advantage of being able to observe serious problems in the oral cavity at once because the imaging technique is standardized and the upper and lower teeth can be seen at a glance, so they are useful in estimating age [32].

Therefore, this study confirmed the potential possibility of age estimation using AI in terms of clinical aspects of oral care, as well as forensic medicine, by determining the difference between the actual age and predicted age using panoramic radiographic images, which can be used to evaluate the overall oral condition.

Table 2 lists the root mean square errors and accuracies according to the teeth status, as the dataset used in the experiment has three types of teeth: healthy, treated (except implant), and with implants. The root mean square errors (RMSE) of the different teeth status were between 6.4 and 8.3. The performance with the healthy teeth was the best with an accuracy of 96.453% and an RMSE age of 6.4563. This indicates the proposed model was not biased toward the implants or treated teeth when estimating the ages of the teeth images.

Although the ages of the subjects could be predicted from their dental panoramic images to within 15 years of their actual ages in approximately 95% of the cases, errors greater than 25 years comprised 0.419% of the cases. From an analysis of images that resulted in errors, dental prosthetic materials, such as implants, and periodontal diseases requiring endodontic treatments were found to be the major causes of these errors in dental age prediction from dental radiographs. The presence of several implants or endodontic treatments in dental panoramic images can cause misclassification into older or younger age groups. Furthermore, images with no teeth or panoramic images with almost all teeth implanted were predicted to be of younger people in this study (Figure 12). This is thought to be due to a lack of training data as a result of the significantly low data for edentulous jaws, as well as misrecognizing implants as natural teeth.

Table 3 lists the comparison of the proposed method to the conventional studies in the aspect of the accuracies and datasets. The classification accuracy with a ±15 year tolerance is higher than the most conventional results [17,33,34,35,36,37] except Mualla et al’s study [38], but the accuracy of the proposed method with a ±5 tolerance is relatively lower. One of the expected reasons for this result is the type of input data. Different from the previous experiments, the utilized images in this study have approximate age information only. For example, a tooth image of 29 years may be confused with the age of 30 easily, but it was considered as a 6-year error (which is categorized as an error within ±15) in our experiments because the dataset had no precise information.

Nevertheless, in this study, it was confirmed that the accuracy of ±15 was very high at 99.521%. This is a very high result compared to the accuracy of 56.5–69.8% when confirmed with ±10 years, which is the age error range commonly used in forensic dentistry [39]. However, since there is a difference in the age error range from this study, care should be taken in simple comparison.

This study has full potential because age estimation using CNN can obtain more accurate results than traditional manual methods that are labor-intensive and time-consuming. Age estimation serves a key function in forensic anthropology and evidence, particularly in criminal investigations and disasters [3,4]. Despite rapid advances in DNA sequencing technology, age estimation using DNA methods is not generally available [40]. Therefore, age estimation using CNN through oral radiographs is highly likely to be used because it is easy to access. The results of this study contribute to the field of dental care. If the dental age of a person is estimated to be lower than the actual age, that person will be more satisfied with the results and more interested in dental care; however, if the dental age is estimated to be higher than the actual age, the person will be aware of dental care and have regular oral examinations and treatments. Oral health education is essential for maintaining healthy oral conditions and a dental age that matches or is lower than the actual age [41].

Visual educational materials that are easily understood and delivered should be used to promote oral health and healthy oral care habits. Therefore, age estimation through panoramic images of oneself can not only identify one’s own oral health status but can also be provided as educational material for improving oral health. When compared to members of the same age group, one can easily identify which age group one’s oral health status belongs to, which provides motivation to maintain oral health [42]. The main reason for this low accuracy is believed to be the uncertain labeling of age in the data. The precise differences between teeth of similar ages could not be determined because accurate age labels for the training images were not known. Collecting additional data and information by collaborating with dental clinics is planned for additional experiments. A model that can recognize and distinguish normal and abnormal teeth, implants, periodontal diseases, etc. can be developed by overcoming the limitations of this study. In addition, research should be conducted based on the actual ages of the subjects and not the median age values used in this study. Furthermore, such a model can be used in clinical practice to increase the interest of patients in dental care and to compare actual ages with estimated dental ages.

## 5. Conclusions

This paper presented a work to estimate tooth age groups when the precise age information of the tooth is not known. By comparing and analyzing the actual ages of the subjects, the overall recognition accuracy was found to be acceptable. Therefore, this study has the potential to be used as oral health education material using the difference between the actual age and the predicted age through AI in dental clinics. This has a very high clinical significance because it will play a positive role in motivating patients in terms of oral health.

## Figures and Tables

**Figure 1 healthcare-11-01068-f001:**
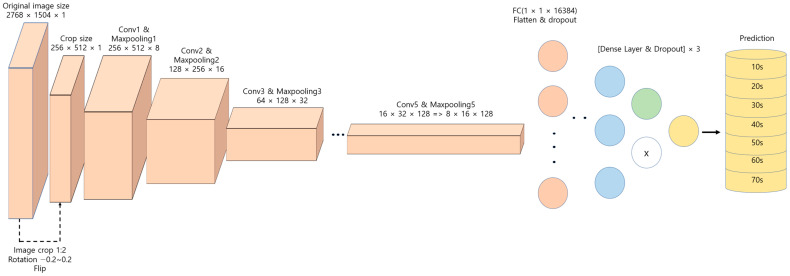
Network architecture of the proposed AI model.

**Figure 2 healthcare-11-01068-f002:**
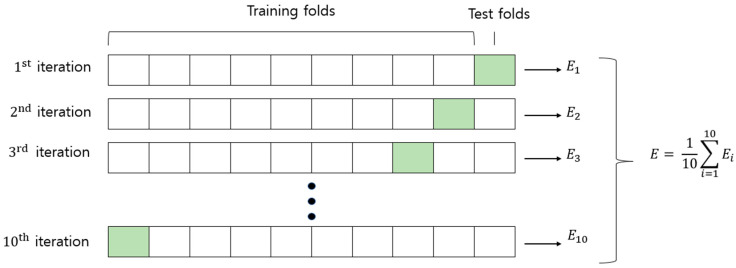
Ten-fold cross-validation contributing to the accuracy and error in this study.

**Figure 3 healthcare-11-01068-f003:**
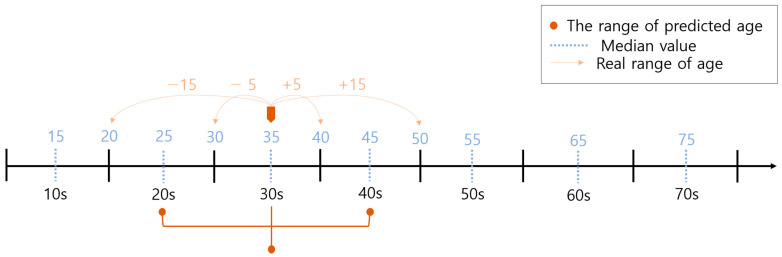
Setting the median values and the range of the predicted dental age.

**Figure 4 healthcare-11-01068-f004:**
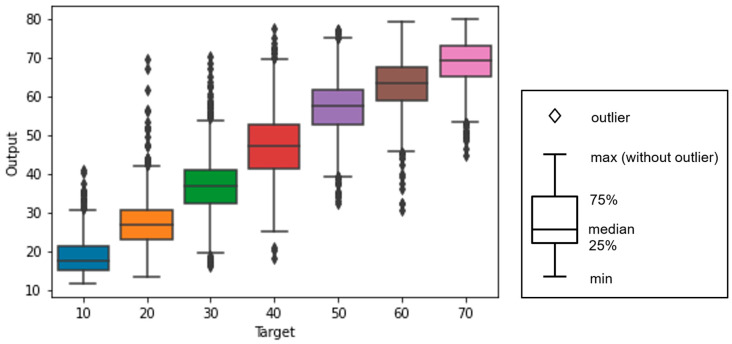
Boxplot of predicted ages according to their actual age group: the colored boxes indicate 25–75% of the age values predicted by the proposed model.

**Figure 5 healthcare-11-01068-f005:**
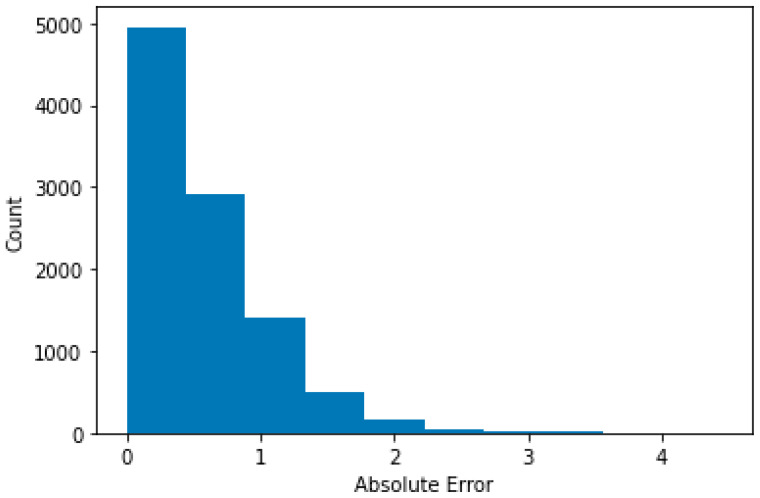
Number of images with errors according to the absolute error in predicted ages.

**Figure 6 healthcare-11-01068-f006:**
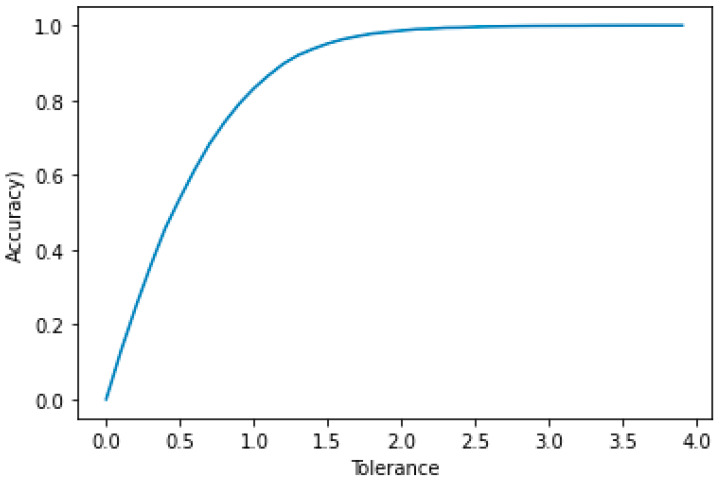
Graph showing the accuracy of the predicted dental age for each tolerance.

**Figure 7 healthcare-11-01068-f007:**
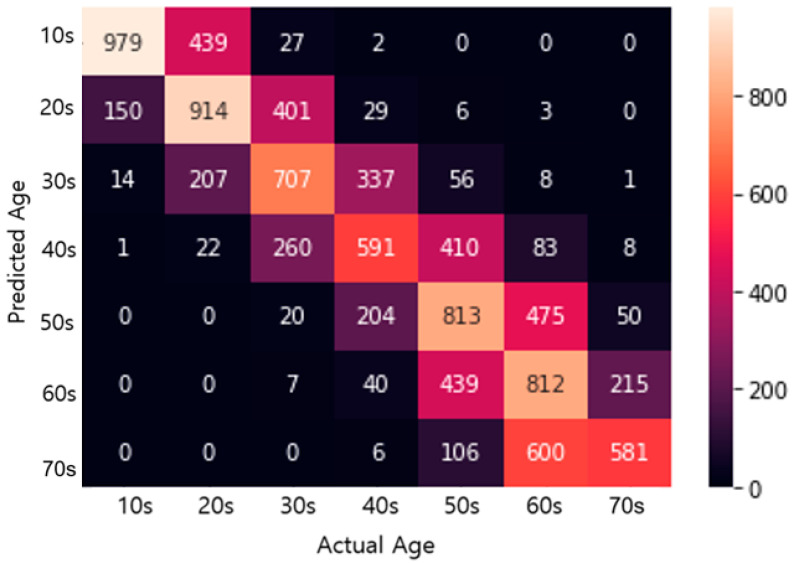
Confusion matrix with actual and predicted ages.

**Figure 8 healthcare-11-01068-f008:**
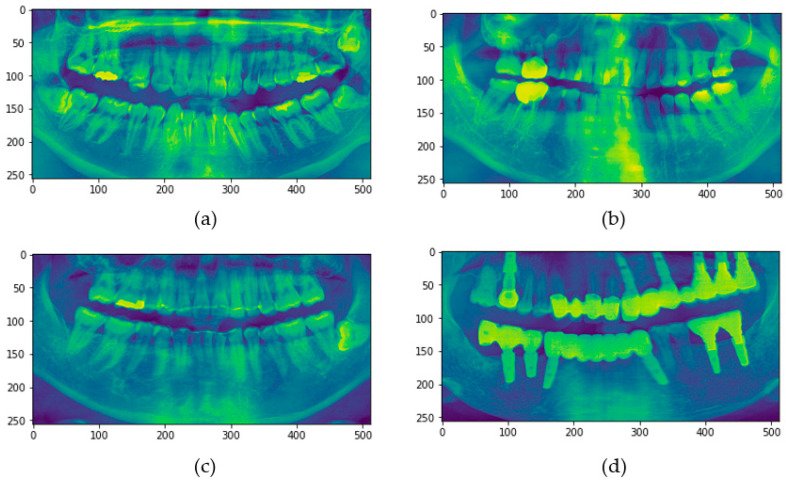
Dental radiographs of persons whose dental ages were successfully predicted using panoramic radiography to within ±5 years: (**a**) actual age is in the 20s, predicted dental age is 24.41; (**b**) actual age is in the 60s, predicted dental age is 60.88; (**c**) actual age is in the 30s, predicted dental age is 32.67; and (**d**) actual age is in the 60s, predicted dental age is 61.85.

**Figure 9 healthcare-11-01068-f009:**
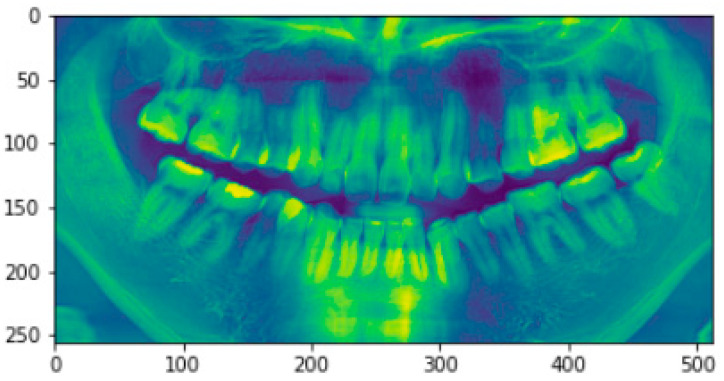
Example of a misclassified image with an error of more than five years; the actual age was in the 20s, but the predicted age was 30.61.

**Figure 10 healthcare-11-01068-f010:**
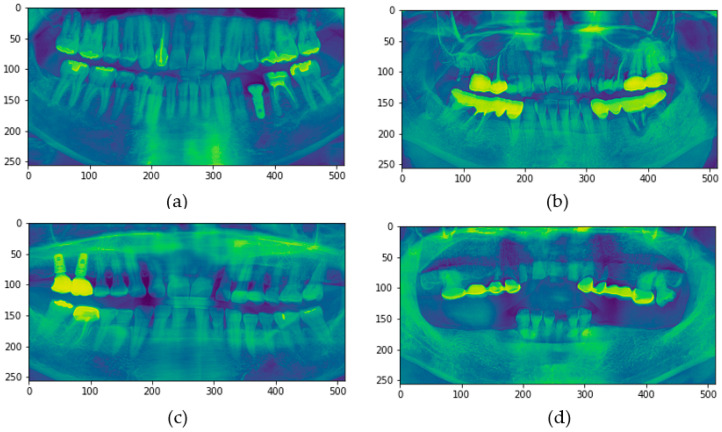
Dental radiographs of persons whose dental ages were misclassified by more than 15 years: (**a**) actual age is in the 50s, predicted dental age is 38.31; (**b**) actual age is in the 30s, predicted dental age is 55.07; (**c**) actual age is in the 70s, predicted dental age is 59.41; and (**d**) actual age is in the 30s, predicted dental age is 70.09.

**Figure 11 healthcare-11-01068-f011:**
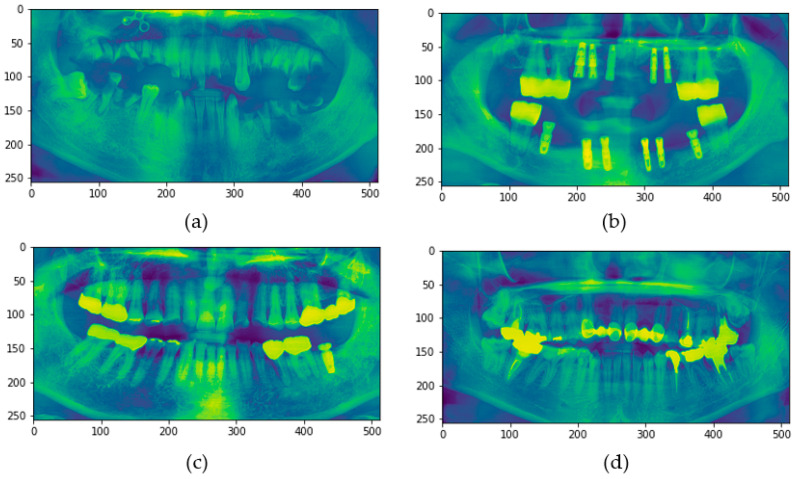
Dental radiographs of persons whose dental ages were misclassified by more than 25 years: (**a**) actual age is in the 20s, predicted dental age is 67.22; (**b**) actual age is in the 20s, predicted dental age is 69.52; (**c**) actual age is in the 20s, predicted dental age is 61.59; and (**d**) actual age is in the 20s, predicted dental age is 51.78.

**Figure 12 healthcare-11-01068-f012:**
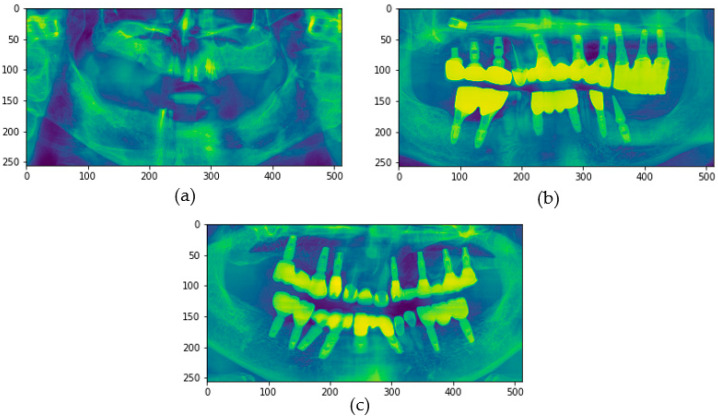
Dental radiographs of persons with dental implants whose dental ages were predicted to be lower: (**a**) actual age is in the 70s, predicted dental age is 55.49; (**b**) actual age is in the 70s, predicted dental age is 52.46; and (**c**) actual age is in the 70s, predicted dental age is 57.99.

**Table 1 healthcare-11-01068-t001:** Table of accuracies according to tolerances.

Tolerance (Years)	Accuracy (%)	Predicted Range (Years)
±5	53.846	Equal to the median value
±15	95.121	±10
±25	99.581	±20

**Table 2 healthcare-11-01068-t002:** Root mean square errors and accuracies according to teeth status.

Teeth Status	RMSE (Age)	Accuracy (%)		
		±5	±15	±25
All	7.4598	53.846	95.121	99.581
Healthy teeth	6.4563	65.199	96.453	99.817
Treated (except implant)	7.3094	53.596	95.663	99.625
With implant	8.2653	47.684	93.250	99.354

**Table 3 healthcare-11-01068-t003:** Accuracy comparison to the recent results for tooth age estimation.

Ref. No.	Age Range in Dataset	Input Image Type	Age Info.	No. Age Group	No. Images	Performance
[33]	1–17	Whole image	Precise age info.	5	456	81.83 (%)
[34]	15–23	Specific tooth image		5	1000	83.25%
[35]	5–24	Whole image		2	10,257	95.9 (%)
[20]	0–60+	Specific tooth image		3	1586	90.37 (%)
[38]	0–70	Whole image		8	1429	98.8 (%)
[36]	19–90	Whole image		(regression)	4035	$0.84\;(R^{2}$)
[37]	0–93	Whole image		(regression)	27,957	MAE: 1.64 years
[16]	4.5–89	Whole image		(regression)	2289	$0.90\;(R^{2}$)
Proposed	10–80	Whole image	Age group info. only	7	10,023	±5: 53.846 (%) ±15: 95.121 (%) ±25: 99.581 (%)

## Data Availability

Not applicable.

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
