# Peer review of "Age Group Classification of Dental Radiography without Precise Age Information Using Convolutional Neural Networks"

_healthcare, 2023, doi:10.3390/healthcare11081068_

Round 1

Reviewer 1 Report

Dear authors, I wish to congratulate you on your excellent work. This paper is very well-developed and interesting to the reader. There are only a few issues to address. Thank you

Line 68-73: the last sentence of the introduction usually refers to the aim of the study. Please report here only the aim. The last sentence is the conclusion of this study, which must be moved to the conclusion section.

Please explain better fig. 1 in the main text

In the material and methods section: which specific features of each panoramic are considered to give the patient the estimated age? The number of teeth? The presence of implants? The height and consistency of maxillary and mandibular bone? The width of the maxillary sinus? Please explain better what CNN considered.

Conclusions may be reorganized, moving the results and limitations of the study respectively in the results and discussion sections. In this section, authors should only have to explain the real impact of their work to the scientific community with possible clinical implications and future perspectives which may arise from its publication.

Reviewer 2 Report

Dear Authors,

I present my comments

Introduction:

it is quite unclear how the age of the teeth was determined.At the end of this section, there is a summary of the results, but the aim of the research is missing.

Materials and Methods:

There is no information on the parameters of the examination, the apparatus used and the technique of taking X-ray images. It is quite unclear how the age of the teeth was determined. A network model  designed to estimate the age of the teeth should be more precisely described.Figure 1 is incomprehensible, it should be described. The research design is not appropriate, because the condition of  oral health  depends on many factors (general health, hygiene habits, regular visits to the dentist, etc.). Moreover, the precise differences between teeth of similar ages could not be determined because accurate age labels for the training images were not known.

Results:

The obtained results are correctly described, but they are burdened with a large error, which makes them of little use. The accuracy with a tolerance of ±5 years was low (54.574%). An accuracy of 95.11% was achieved when the error tolerance was set to 15 years. Such a large tolerance is unacceptable.

Discussion:

The discussion is pretty poor. The results are unnecessary in this section (line 218-229).

Conclusions:

The conclusions highlight the imperfection of the research conducted.

Reviewer 3 Report

Dear authors, this study proposed to evaluate a deep neural network, as a method to estimate tooth ages when the precise age information is not given.

This study presents several flaws that need to be considered.

 Abstract: ‘AI’, an appropriate information about this abbreviation needs to be provided.

Keywords: the following keywords are not Mesh Terms “age group classification; dental age; CNN”.

Line 71-73: Study results should not be mentioned in the Introduction to the article. Consider removing them.

Line 118-119: “The figure shows that most of the predicted ages were accurate.” How did you arrive at that conclusion? Was any statistical test applied?

Figure 4: Improve your legend with all information about the boxplot.

Line 125: “…and most of the errors…” How many?

Line 139-140: “Figure 7 shows confusion matrix of the estimation results. Clearly, the most confusion occurred…” What is the meaning of classifying or qualifying this parameter? What is the reason for presenting this figure? What does this figure bring of relevant information in relation to the others analyzes carried out?

Line 247-261: no scientific literature was used.

Line 247-252: This is overestimated information without proven scientific basis in the references.

Line 222: “…endodontic treatments were found to be the major causes of errors in dental age prediction…”; Line 223: “the presence of several implants or endodontic treatments in dental panoramic imagens can cause misclassification…”; Line 226-228: “This is thought to be due to a lack of training data as a result of the significantly low data for edentulous jaws, as well as misrecognizing implants as natural teeth.” These artifacts that led to the error should be taken into account in the validation of the method and have sought modifications in the analysis in order to allow a more accurate analysis. I consider this to be the major problem in this study, which devalues the accuracy of the method and compromises the objective of the study.

Reviewer 4 Report

Dear Authors,

your work is an interesting paper focused on the progress in the field of dental-related age estimation, however the readability is not very smooth. I recommend that you improve the introduction perhaps by adding some other methods already used in the staging of dental age (doi: 10.3390/ijerph191610454). Subsequently, I suggest you better specify what is the aim of your and finallyrelocate the limitations of your study from the conclusion to the end of the discussion paragraph.

Round 2

Reviewer 2 Report

Dear Authors,

The corrections made significantly increased the value of the work. The article may be published in present form.

Reviewer 3 Report

Dear authors, thank you for your considerations.